# MACK: Mismodeling Addressed with Contrastive Knowledge

**Liam R. Sheldon**[1*], **Dylan S. Rankin**[2†], **Philip Harris**[1]

**1** Massachusetts Institute of Technology, Cambridge, MA, 02139
**2** University of Pennsylvania, Philadelphia, PA 19104

★ lsheldon@mit.edu ,    † dsrankin@sas.upenn.edu

## Abstract

The use of machine learning methods in high energy physics typically relies on large volumes of precise simulation for training. As machine learning models become more complex they can become increasingly sensitive to differences between this simulation and the real data collected by experiments. We present a generic methodology based on contrastive learning which is able to greatly mitigate this negative effect. Crucially, the method does not require prior knowledge of the specifics of the mismodeling. While we demonstrate the efficacy of this technique using the task of jet-tagging at the Large Hadron Collider, it is applicable to a wide array of different tasks both in and out of the field of high energy physics.

# 1 Introduction

The use of machine learning (ML) in high energy physics (HEP) has grown rapidly in the last decade. This can be attributed to multiple factors, among them the availability of large volumes of accurate simulated data and the complexity of the data. While the early usage of ML focused on problems of limited scope using small sets of input variables, the rise of deep neural networks and the availability of powerful computing have pushed the field towards more advanced model architectures and more complex, low-level inputs [1–4]. This has greatly improved the performance of these algorithms, and in many cases drastically improved the quality of physics results [5–8]. One feature that is common across nearly all of these applications is that models are trained using simulated data. While the accuracy of the simulations is quite high in many cases, the high dimensionality of the data can make it difficult to understand exactly where differences exist and the degree to which they may affect the performance of a given algorithm. This is particularly true as ML models become increasingly complex.

Despite the high accuracy of simulated data in many cases, the differences in ML model performance when applied to simulated and real data can be large. This is due to mismodeling in the simulation that can be subtle and/or difficult to correct. Although this issue is extremely prevalent across the field, there exist few methods to minimize these differences in algorithm performance and most of these existing methods cannot be effectively used for the large, complex models that are becoming increasingly popular.

In this work we present a generic method that is capable of significantly reducing differences in the performance of ML models when applied to simulated and real data. The method is based on techniques from the realm of contrastive learning and is able to construct expressive representations of the data that are minimally sensitive to mismodeling in the simulated data. It does not rely on prior knowledge of the sources of mismodeling or variables which may be mismodeled. The representations can be built using complex architectures and can therefore harness the full power of modern ML.

This paper is structured as follows: Section 2 provides context by describing work related to our method, which is itself described in Section 3. Section 4 provides details on the studies we have performed and results are presented in Section 5 that demonstrate the effectiveness of our method. Finally, Section 6 summarizes this work and provides an outlook for its application to existing and future problems in HEP.

# 2 Related Work

This work relies on many existing tools and ideas in the fields of HEP and ML. Our work focuses on the jet-tagging problem as an example of a problem with significant complexity. Broadly speaking this problem involves using the properties of a jet to determine information about the initial particle that produced the jet. Past work has demonstrated a range of viable architectures, with graph- and attention-based networks that use jet constituents as inputs shown to be the most effective [1, 2]. Our method relies on the presence of a distance metric between events from simulation and data. For this work, we utilize the Energy Mover's Distance (EMD) [9–11] and make use of the package developed in Ref. [12].

The method we propose takes significant inspiration from self-supervised contrastive learning. In particular, we use the Variance-Invariance-Covariance Regularization (VICReg) method from Ref. [13]. We also take inspiration from other contrastive learning methods [14–16]. In particular, the method from Ref. [14] was adapted for jet-tagging in Ref. [17], and we study some of the same augmentations proposed there. We also note that other contrastive learning methods have recently been proposed for the purpose of minimizing differences between ML

performance in data and simulation in Ref. [18, 19].

## 3   Method Description

Our method uses contrastive learning to inject physics knowledge into the representations of data. The goal of contrastive learning in this context is to instruct machine learning algorithms to represent the same physical phenomena in similar ways despite their outward differences. Contrastive learning, at a fundamental level, works by examining pairs of elements and determining whether the two elements in the pair should be represented similarly or differently. It is common to refer to these as positive or negative pairs, respectively. In standard contrastive learning implementations, positive pairs are created by simply augmenting one element of data to create a pair from the original and the augmentation. Negative pairs are constructed by taking two different elements (or augmented versions of these different elements). However, this requires that the augmentations can mimic differences to which one desires the model to be insensitive. In our case, we want a model to become desensitized to differences between simulation and data but we cannot construct an augmentation that transforms actual data to simulation. Instead, the central idea of our method is to construct pairs from actual data and simulation and then label them as positive or negative based on a distance metric. We find that EMD is an effective metric for this purpose.

In the studies below the physical phenomena are jets of particles from simulated $pp$ collisions at the ATLAS [20] or CMS [21] detectors at the Large Hadron Collider [22] and the task is to identify the type of particle from which the jet originated. One of the datasets is taken to represent actual data collected by a detector while the other is taken to represent simulation that would be available and is a reasonable approximation of the data, but is not perfect. In practice, our simulation would be labeled while our actual data would not. Practically, this means that methods can use truth labels for simulation during training but they must remain agnostic to truth labels for data. Thus, we can use the truth labels for the actual data only when assessing the performance and stability of models and not at any point during training. For simplicity we refer to these datasets as the "nominal" and "alternate" datasets, with nominal representing simulation and alternate representing data.

In a typical HEP data analysis, one would train a supervised classifier to differentiate signal and background using simulated events only. This model would then be applied to both data and simulated events representative of that data. Because the model can perform differently on data and simulated events, a ratio of the model performance on these samples is typically computed in a dedicated control region and then used to correct simulated events. Particularly in cases where the differences between data and simulation are large, this correction necessitates an additional source of uncertainty which can negatively impact the sensitivity of the analysis. Even in cases where control region measurements allow one to reduce the uncertainty on this correction, large corrections are indications that the model has learned features specific only to simulated events and as a result the performance in data is likely sub-optimal. Our method seeks to allow for training models that perform more similarly on simulated and real data, require smaller systematic uncertainties, and therefore can produce more sensitive physics results.

The general form of our contrastive model is as follows, depicted in Fig. 1. Our model consists of a "siamese network" comprised of two sub-networks: a featurizer and a classifier network. The featurizer and projector network each separate share weights between the legs of the siamese network. To start, each element of a pair goes into the featurizer which creates a representation $L$ and then through a projector which creates a representation $P$ for each input. These representations are trained through contrastive learning methods which compare

the two representations $P$ and $P'$ for a given pair. After the featurizer is trained we apply it to the nominal dataset and train a supervised model using the labels. In the studies below we consider only the task of binary classification, but it is important to stress that the method allows for any type of model, including multi-class classifiers or regression models, to be trained. Additionally, while our main goal isn't to make our model immune to more traditional symmetries in collider physics, our method also allows for the addition of these sorts of symmetries. As such we do incorporate other augmentations on top of our base method, in particular those for rotations and smearing from Ref. [17]. These augmentations can be applied to either the nominal or alternate datasets (or both) once the pairs have been constructed. Throughout we denote our method as Mismodeling Addressed with Contrastive Knowledge (MACK).

In this work we have only considered cases where the featurizer is made identical for simulated and real data. However, this is not enforced by the method we have described above. Allowing different featurizers for simulated and real data could allow the model to better tune the separate featurizers. Ideally, this flexibility would produce output features that are even more similar but it's also possible that it would result in larger differences or unforeseen impacts on downstream tasks. We leave further exploration of this possibility to future work.

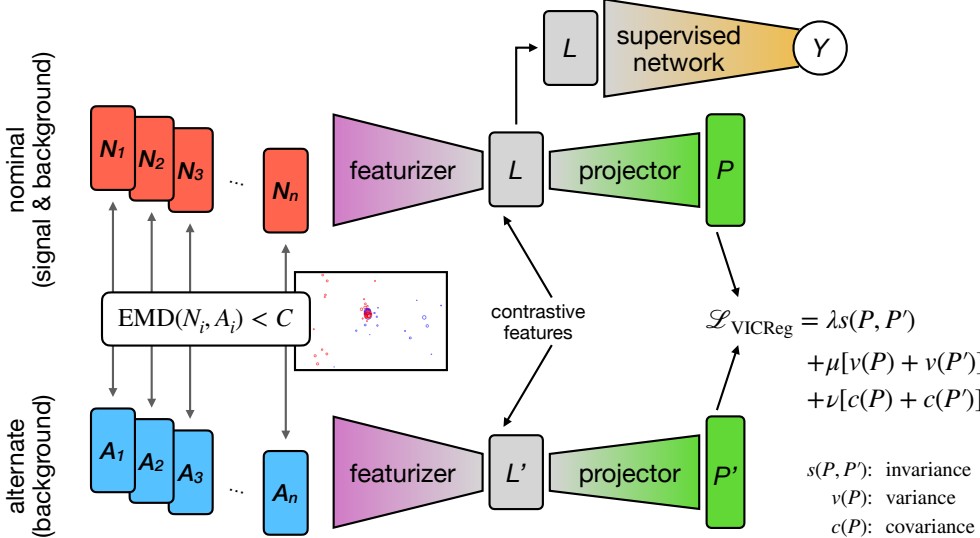

Figure 1: Sketch of the MACK method. Samples from the nominal and alternate datasets are paired such that the EMD between each pair is less than some cutoff $C$. These pairs are used to train a featurizer and a projector network without labels using the VICReg loss ($\mathscr{L}_{\mathrm{VICReg}}$) applied to the outputs of the projectors ($P$ and $P'$). The outputs of the featurizer when applied to the nominal dataset ($L$) are then used, along with labels, to train a desired supervised network.

## 4  Study Details

We study two datasets both focused on the task of boosted jet-tagging at the LHC. The first is a realistic example of a task that would require our contrastive method and allows us to very effectively study and demonstrate the power of the MACK method. The second is less realistic but allows us to test MACK in the presence of extreme degradation in performance between

datasets.

## 4.1 Datasets

In the first study, we consider the task of binary classification of jets originating either from an energetic light quark or gluon through QCD multijet production or from the decay of a hypothetical boosted $Z'$ boson with a mass of 50 GeV decaying to two quarks ($Z' \to q\bar{q}$). This represents a task at the LHC that requires the use of powerful ML methods and might suffer from differences between simulation and data. We refer to these as the *realistic datasets*.

QCD multijet and $Z'$ samples are generated using MADGRAPH5_AMC@NLO v2.6.0 [23] at $\sqrt{s} = 13$ TeV, showered with PYTHIA v8.212 [24], and reconstructed using DELPHES v3.5.0 [25] with a CMS detector-like geometry. The $Z'$ sample is generated using the FeynRules-based DMsimp package [26–29], with the $Z'$ recoiling against a photon to provide the necessary boost. The QCD multijet sample is generated such that the $p_T$ of the leading parton is similar to that of the $Z'$.

Jets are clustered from the DELPHES E-Flow candidates using the anti-$k_t$ algorithm [30] with a distance parameter $R = 0.8$. The softdrop grooming procedure [31] with $\beta = 0$ and $z_{cut} = 0.1$ is applied to jets to compute the softdrop mass ($m_{SD}$). In each event, we select a single jet and use the 30 highest $p_T$ particles in the jet as input to our networks. For the QCD multijet sample, we select the jet with the highest transverse momentum $p_T$. For the $Z'$ sample we select only the jet in each event that is within $\Delta R < 0.4$ of the true $Z'$ boson, where $\Delta R = \sqrt{(\eta_{jet} - \eta_{Z'})^2 + (\phi_{jet} - \phi_{Z'})^2}$. We require jets to have $p_T > 200$ GeV and $m_{SD} > 20$ GeV. For each particle we compute four features: $p_T^{rel} = p_{T,particle}/p_{T,jet}$, $\eta^{rel} = \eta_{particle} - \eta_{jet}$, $\phi^{rel} = \phi_{particle} - \phi_{jet}$, and $q$, where $q$ denotes the particle's charge.

For both the QCD multijet and $Z'$ samples we generate two distinct sets of events. In one set we use TuneCUETP8M1 [32], while in the second we use TuneCUEP8M2T4 [33]. We generate roughly one million events per sample per set. We treat the first dataset as the nominal and the second dataset as the alternate.

The second study uses the JetNet dataset [34], which consists of the $p_T^{rel}$, $\eta^{rel}$, and $\phi^{rel}$ of the 30 highest $p_T^{rel}$ particle constituents for jets originating from light quarks (q), gluons (g), W and Z bosons, and top quarks (t). The dataset contains approximately 170,000 jets for each class. We require jets to have $500 < p_T < 1500$ GeV and $m > 20$ GeV. We take jets originating from light quarks and W bosons as the nominal dataset and jets originating from gluons and Z bosons as the alternate dataset. That is, for the supervised case we train a binary classifier to differentiate between light quarks and W bosons (nominal), and then test this classifier on gluons and Z bosons (alternate). Unsurprisingly, given the substantial differences between light quarks and gluons and W and Z bosons, the difference in performance is significant. Because there is no particle ID information present in the JetNet dataset inputs, the primary difference between the jets originating from W and Z bosons is in the jet mass.

## 4.2 Supervised Model Design

As a baseline we construct a supervised model based on the architecture from Ref. [35], using the 30 particles and three or four features per particle as given above. This architecture is a fully-connected graph neural network (GNN) with three main sub-networks: a node→edge network ($f_R$), a node feature extractor ($f_O$), and a final classifier block ($\phi_C$). In our specific implementation, $f_R$ consists of three dense layers of size 64, 32, and 32, $f_O$ consists of three dense layers of size 64, 32, and 32, and $\phi_C$ consists of three hidden dense layers of size 64, 32, and 8, with a single output node. All layers use a ReLU activation function except for the final layer of $\phi_C$ which uses a sigmoid activation function. We note that while more advanced

network architectures exist, the GNN architecture we employ is sufficiently complex to allow for substantial over-reliance on the differences between the nominal and alternate datasets.

The model is implemented in the Keras [36] framework. Using only the nominal dataset we train this network using the Adam optimizer [37] for 100 epochs using a binary cross entropy loss function and a batch size of 512. We employ an early stopping criteria using a patience of 10 epochs. We split the data 80/10/10 to produce training/validation/testing sets.

### 4.3 MACK Model Design

For the MACK featurizer network we use the same basic GNN architecture as the supervised network presented above. The only difference is that $\phi_C$ now consists of four layers, two of size 128 and two of size 64, all of which use a ReLU activation function. The MACK projector network is comprised of two dense layers both of size 64, the first using a ReLU and the second a linear activation function. The use of a projector is motivated by the studies in Ref. [14] which demonstrate that the presence of a non-linear projector can improve the performance of tasks when trained on the pre-projector representation. The size of the projector is found not to impact performance significantly. The combination of the GNN architecture along with the projector is used as the legs of the siamese network described in Section 4.

We consider the loss proposed in the VICReg method [13] applied to the outputs of the projector ($P$ and $P'$). The $\lambda$, $\mu$, and $\nu$ weights in the VICReg loss are taken as-is from the reference and set to 25, 25, and 1, respectively. Other contrastive losses are possible with MACK but were not studied.

We train the featurizer and projector networks using this loss and the Adam optimizer for 100 epochs using a batch size of 1024. We employ an early stopping criteria using a patience of 10 epochs. Using the representations constructed by the featurizer network as input, we then train a simple dense neural network binary classifier using the nominal dataset. This network has 3 hidden layers of size 64, 32, and 8, and one output node. The hidden layers use ReLU as the activation function and the output node uses a sigmoid as the activation function. This classifier model is trained using the Adam optimizer, a batch size of 1024, and uses an early stopping criteria with a patience of 10 epochs. In initial tests of MACK models a pair consisted of one jet from the nominal dataset and one from the alternate dataset, with these pairs created by matching jets using EMD. Pairs of jets with a non-normalized pairwise EMD less than 75 GeV were labeled as positive pairs in the realistic datasets. This threshold was determined empirically such that the fraction of pairs with EMD less than this value was approximately 1/4, but minimal difference in performance was observed for other thresholds close to this value. The threshold for the JetNet dataset was chosen to achieve a similar fraction of positive pairs. During early studies it was observed that the addition of augmentations for rotations and smearing from Ref. [17] improved performance. This observation is consistent with previous studies in Refs. [14,17] that certain augmentations are much more effective than others and that combinations of augmentations are more effective than individual augmentations. A comparison of various pairing schemes is shown in Fig. 2. Pairs are constructed using both EMD and augmentations for MACK models in the following results. Scans of the featurizer, projector, and classifier networks showed minimal dependence on their size.

## 5 Results

For both datasets, we compare the performance of MACK to the supervised benchmark model. Performance on both the nominal, as well as alternate, datasets is considered, as well as the differences in performance on these two samples.

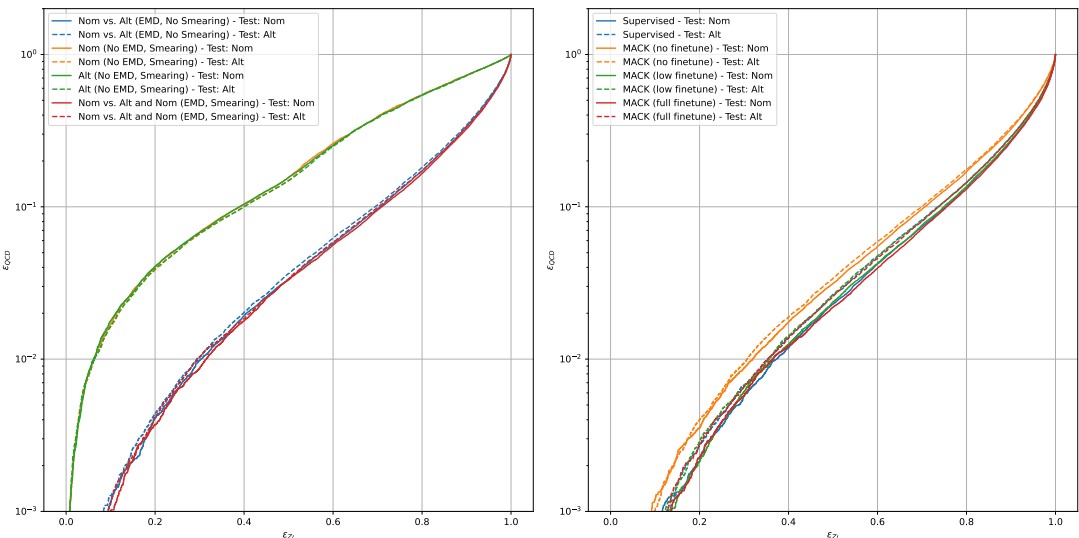

Figure 2: ROC curves for the supervised model and MACK with different levels of fine-tuning and augmentations, evaluated on the realistic datasets ($Z' \rightarrow q\bar{q}$ vs. QCD). Left: Comparison of different contrastive models using EMD-based pairings and/or particle augmentations. A clear improvement is seen with EMD-based pairing (blue) over augmentations alone (orange or green), with a small additional improvement from the combination of EMD-based pairing and augmentations (red). Right: Comparison of the supervised model (blue) and MACK with different levels of fine-tuning (orange, green, red), evaluated on the realistic datasets. Even small amounts of fine-tuning are capable of producing MACK models with performance similar to or better than a supervised model.

## 5.1   Realistic dataset: $Z' \rightarrow q\bar{q}$ vs. QCD

We find that MACK is able to reduce variance in performance between models when they are tested on the nominal and alternate datasets. To help quantify this effect we define the metric fractional change in efficiency, $\Delta\epsilon/\epsilon$, as follows: using the nominal dataset we compute the classifier output threshold required to produce a given QCD ($Z'$) efficiency, and then compute the fractional difference in $Z'$ (QCD) efficiency when that threshold is applied to the nominal and alternate datasets. This procedure mimics how an ML classifier is used in typical HEP analyses and $\Delta\epsilon/\epsilon$ therefore represents a scale factor that would need to be applied to account for data/simulation differences. As shown in Fig. 2 and Fig. 3, the differences in performance for MACK when tested on nominal data and alternate data are much smaller than the differences in performance for the supervised model when tested on nominal and alternate data. This implies that MACK is able to better learn the features of jets that are similar between the nominal and alternate datasets than the supervised model, and ignore those features that are different. We note that this increased stability with respect to differences in the two datasets comes at the cost of overall performance, as the supervised model performs better on both datasets than MACK.

    In order to address the overall difference in performance between MACK and the supervised model we employ a fine-tuning procedure. While baseline MACK uses the representations from the featurizer directly, fine-tuned MACK allows the weights in the featurizer network to be updated during the classifier training (full fine-tuning). This allows the featurizer to use more features relevant to the specific classification task and therefore improves the performance on both the nominal and alternate datasets. However, we also find that fine-

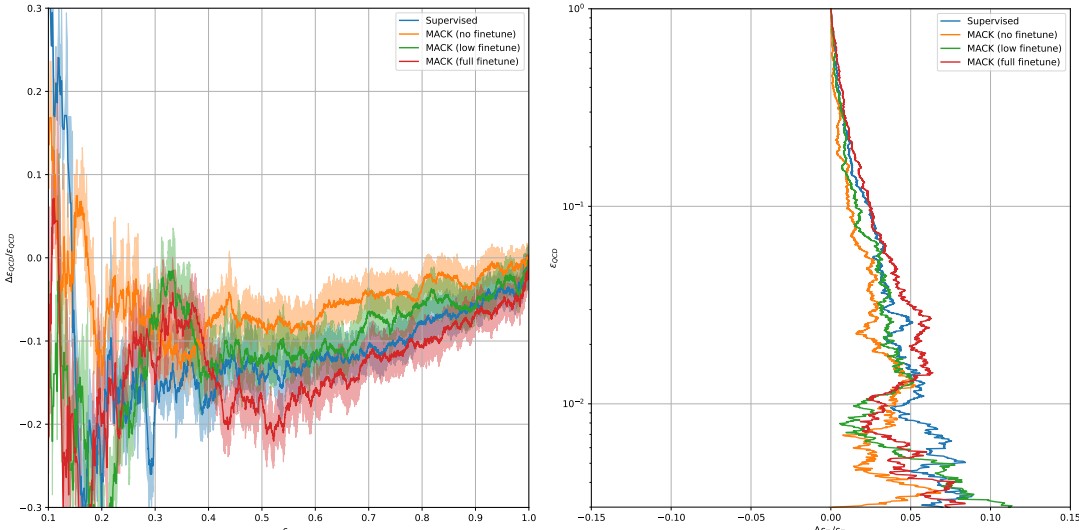

Figure 3: Comparisons of the performance for the supervised model (blue) and MACK with different levels of fine-tuning (orange, green, red), evaluated on the realistic datasets. MACK with no fine-tuning shows a fractional change half that of a supervised model. Left: Fractional change in QCD efficiency ($\Delta\epsilon_{QCD}/\epsilon_{QCD}$) at a fixed $Z'$ efficiency. Right: Fractional change in $Z'$ efficiency ($\Delta\epsilon_{Z'}/\epsilon_{Z'}$) at a fixed QCD efficiency.

tuned featurizers are more prone to exploiting specifics of the nominal dataset since they are fine-tuned using only this dataset. By allowing the featurizer weights to be updated for a small number of epochs and then frozen for the remainder of the classifier training (low fine-tuning) we are able to achieve performance near that of the supervised model with reduced dataset dependence. All our results below allow use 5 epochs to fine-tune MACK in the low fine-tuning mode. This is chosen to balance the advantages in performance and disadvantages in stability from fine-tuning.

Table 1 summarizes the results of our studies on the realistic datasets. In general, we find that the best performance on the nominal dataset is achieved by fully fine-tuned MACK, while the smallest difference in performance between the nominal and alternate datasets is achieved by base MACK (no fine-tuning). Low fine-tuned MACK strikes a balance between these two and achieves comparable results to the supervised model with reduced differences between the nominal and alternate datasets.

## 5.2 JetNet dataset: $q(g)$ vs. $W(Z)$

As in the studies above we find that MACK can be successfully applied to the task of separation of light quarks and gluons from W and Z bosons. In this case, the differences between the nominal ($q$ and W) and alternate ($g$ and Z) datasets are extreme and are not typical of those expected between data and simulation in most situations. However, this large difference provides a unique test of the impact of MACK with different settings. As shown in Fig. 4, the differences in performance for MACK when tested on nominal data and alternate data are again significantly smaller than the differences in performance for the supervised model when tested on nominal and alternate data. While the difference between the two datasets is still quite large in this case it is nearly an order of magnitude less than for the supervised model. This increased stability comes at the cost of overall performance on the nominal dataset when

| Model | Supervised | MACK | | |
| --- | --- | --- | --- | --- |
| | | No Finetune | Low Finetune | Full Finetune |
| Nominal AUC | 0.911 | 0.894 | 0.911 | **0.913** |
| Alternate AUC | **0.906** | 0.891 | **0.906** | **0.906** |
| $\epsilon(\text{QCD}) @ \epsilon(Z') = 0.3$ | **0.005** | 0.008 | 0.006 | 0.006 |
| $\epsilon(\text{QCD}) @ \epsilon(Z') = 0.5$ | 0.022 | 0.031 | 0.023 | **0.021** |
| $\epsilon(Z') @ \epsilon(\text{QCD}) = 0.1$ | 0.744 | 0.698 | 0.742 | **0.747** |
| $\epsilon(Z') @ \epsilon(\text{QCD}) = 0.01$ | **0.289** | 0.232 | 0.280 | 0.278 |
| $|\Delta\epsilon/\epsilon(\text{QCD})| @ \epsilon(Z') = 0.3$ | 0.250 | 0.085 | **0.079** | 0.129 |
| $|\Delta\epsilon/\epsilon(\text{QCD})| @ \epsilon(Z') = 0.5$ | 0.153 | **0.076** | 0.120 | 0.169 |
| $|\Delta\epsilon/\epsilon(Z')| @ \epsilon(\text{QCD}) = 0.1$ | 0.025 | **0.013** | 0.019 | 0.025 |
| $|\Delta\epsilon/\epsilon(Z')| @ \epsilon(\text{QCD}) = 0.01$ | 0.080 | **0.021** | 0.065 | 0.043 |

Table 1: Summary of results for the supervised training and different versions of our method, measured on the realistic datasets. The $\epsilon(QCD)$ and $\epsilon(Z')$ values use the nominal dataset. Bold values indicate the best-performing model based on each metric.

compared to the supervised model. In this case, however, baseline MACK achieves the best performance on the alternate dataset, likely a result of the extreme sample differences and thus large degradation in the performance of the supervised model.

The large differences in the JetNet dataset allow for additional studies of fine-tuning. The MACK models shown in Fig. 4 each start from the same base MACK training and then apply fine-tuning for a different number of epochs. In this case, a single epoch is sufficient to greatly improve the nominal performance. However, we also observe some variance in the performance after training. In particular, fine-tuning for 5 epochs produces a model whose performance is closer to the base MACK model than fine-tuning for 1 epoch. It is clear that fine-tuning is a powerful method for improving contrastive models, and that the specifics of its application require further studies.

# 6 Conclusion

We have introduced a method, MACK, based on contrastive learning, that is able to reduce the sensitivity of ML models to the dataset on which they are trained. MACK achieves this by learning a set of highly expressive features of the inputs that are insensitive to dataset differences. These features can then be used for downstream tasks; in our case we consider two binary classification tasks, but note that the method can easily be applied to other tasks such as regression. Crucially, MACK does not require a priori knowledge about the nature or source of differences between datasets. This makes it possible to apply MACK in a very wide range of problems. Although MACK is designed specifically with HEP tasks in mind, the method is generic enough to be used in other fields.

In tests with datasets mimicking differences expected between simulated and real data collected at the LHC, we find that the performance of MACK is more stable than a supervised classifier with a similar architecture. This stability comes at some cost in overall performance, but this can be recovered through a fine-tuning procedure that allows the MACK features to be adjusted for the specific downstream task. Allowing the fine-tuning to fully modify the MACK features produces a model that is capable of outperforming even the supervised model, although it also becomes more sensitive to dataset differences as a result.

We also study the performance of MACK under extreme conditions where the differences

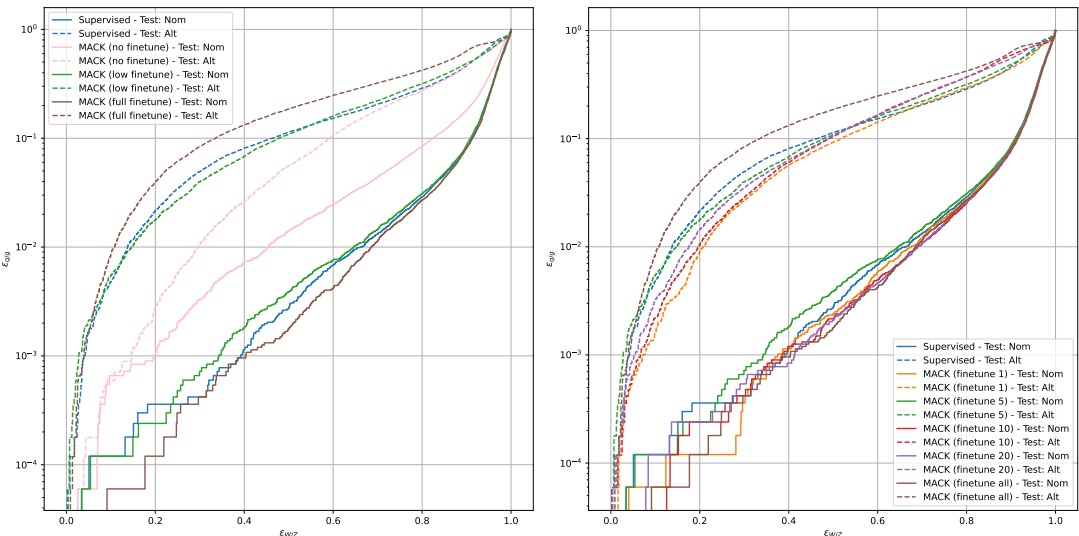

Figure 4: Comparisons of the performance for the supervised model and MACK with different levels of fine-tuning, evaluated on the JetNet dataset. Left: $q(g)$ efficiency as a function of $W(Z)$ efficiency evaluated on the nominal (solid) and alternate (dashed) datasets. The best performance on the alternate dataset is achieved with MACK and on the nominal dataset with MACK after full fine-tuning. Right: Fractional change in $Z'$ efficiency ($\Delta\epsilon_{Z'}/\epsilon_{Z'}$) at a fixed QCD efficiency. Each fine-tuning model is a distinct training starting from the base MACK model. The general trend from fine-tuning for 1 epoch to full fine-tuning is as expected with full fine-tuning achieving the best performance on the nominal dataset and the worst performance on the alternate dataset.

between datasets are large. These tests again show that MACK is capable of reducing dependence on features specific to one dataset and that the use of fine-tuning can produce models that are competitive with or even better than a supervised model. The success of MACK on this dataset strongly suggests that the method is highly robust to the specifics of the dataset differences and that it could be applied to a broad range of tasks. For this dataset, as with the realistic ones, we find fine-tuning to be a very effective tool for improving contrastive models. The exploration of schemes to determine how best to perform this fine-tuning is left to future work.

MACK and other contrastive learning techniques show significant promise for improving the use of ML in HEP. Even without a significant dependence of a given model on the specifics of the training dataset, we find that MACK can be used to improve models either in terms of stability or overall performance. The stability across datasets demonstrated by MACK could also likely be leveraged for other applications in anomaly detection and searches for highly model-dependent new physics. The performance improvements alone suggest that MACK could be used to improve many existing ML tools currently in use in HEP. The application of MACK to even more realistic datasets and even to actual LHC data is an exciting direction for future work.

# Acknowledgements

This method is named for Mack Rankin Sheldon. All work was carried out on the MIT Satori cluster.

**Author contributions** LS wrote most of the code to allow model training, performed all studies in this work, and contributed to parts of the text. DR was responsible for dataset production, code improvements, and wrote most of the text. PH provided initial feedback on the methodology and suggestions for improvement.

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
