# Peer review of "MACK: Mismodeling Addressed with Contrastive Knowledge"

_SciPost Physics_

## Round 1 · Referee Report · Anonymous (Referee 1) · 2024-11-12

Strengths
1- New methodology to reduce simulation bias 2- Realistic test to evaluate the performance
Weaknesses
1- Lack of clarity in the methodology: besides the figure, there's no mention of how the contrastive loss is used, the figure indicated that only background is used in the alternative samples for the featurizer, but the text seems to indicate that signal + background is used. 2- Lack of code for reproducibility or making the realistic sample available
Report
Requested changes
1- As mentioned in the weaknesses, the authors should clarify the motivation for the specific contrastive loss used. Since it's a major portion of thee work, the authors should at least write down the loss in the text and explain each term instead of simply giving a citation. 2- How is the alternate samples constructed? Is it only background as Fig. 1 suggests or is it a mix of both? I would expect that you want the alternate sample to follow as close as possible the training samples, i.e same expected composition and fraction of signal/background events to have a sensible EMD pairing, is this correct? If so, is this an issue with the current implementation? 3-Fig.2 The ROC curves are very hard to read and impossible if you are colorblind. The figure label helps but the legend needs improvement. Have 2 boxes, one that just says that testing on nominal is a full line and testing on alternative samples is a dashed line. In the other box, make it clear and possibly with other colors what do they mean: one line for MACK without augmentations, one for MACK with augmentations, one for trained on nominal aug., one for trained on alternative aug.. 4- In the same figure, I don't understand why training on nominal and evaluating on nominal has such a low performance compared to MACK. Worse, the performance on both nominal and alternative samples seem to be equally bad. Where does the additional performance comes from? Maybe I'm reading the plot wrong, which reinforces my point that the labels aren't clear. 5- Table 1: would be great to add errors to that table to see the stability of MACK across different runs. 6- The extreme example is interesting, but also highly unrealistic. However, I would be curious to see what happens if you keep the training strategy on JetNet as you have, with the alternate samples coming from a mixture of other physics processes, but evaluated the classifier on the same physics processes used for training, but with some modification, either a different tune as you already did in the previous exercise, or different generator. The main reason I ask is because I would like to know what is the impact of the choice of "data" used when calculating the pairs and aligning the embedding. If my data have other processes, or if the fraction of signal and background is different, does MACK hurt or still helps?
Recommendation
Publish (meets expectations and criteria for this Journal)

---

## Round 1 · Referee Report · Anonymous (Referee 2) · 2024-12-17

Strengths
- Novel approach to address mismodelling between data and simulations
- The method itself is agnostic of the subsequent downstream task and thus very general
- Realistic example to evaluate the performance
Weaknesses
- The description of the method is not clear and lacks details
- The presentation of the results and figures is suboptimal
Report
Requested changes
- The method description needs more details about the training procedure. It would greatly benefit from introducing all used loss terms instead of only citing other papers.
- On page 3, in the last paragraph: "...a featurizer and a classifier network. The featurizer and projector network each separate share weights between the legs of the siamese network." In the first sentence you call it a classifier, and then a projector? Is it the same? It needs to be clarified.
- While you cite a paper in Section 4, that the usage of an additional projector after the featurizer is well-motivated, I would suggest adding this line of argumentation to Section 3, where these networks are introduced, and maybe add 2-3 lines of explanation why this is a good idea. Before reading this sentence, I asked myself in Section 3, "OK, but why do you need this additional projector exactly?" So far, it is not clear.
- Also, given that you have multiple networks and subnetworks with different training phases, it is not immediately apparent which (sub)network is trained or kept fixed. This could be summarized with a table or plot.
- If I understand correctly, your supervised network trained on the feature space L differs from the "Supervised Model Design" you mention in Section 4.2, doesn't it? Right now, it is not clear. Parts of the models are introduced in Section 3, and other parts only appear in Section 4. What about streamlining this into a single section, where all networks and classes are properly introduced, and then the introduction of the dataset becomes part of Section 5, which is about the actual application? This would make the storyline much more straightforward. Moreover, this structure would emphasize again that the MACK model itself is independent of the actual downstream task.
- The label and legend sizes of your Figures 2, 3, and 4 are currently tough to read, and the plot makes it very difficult to understand what is happening. This could be illustrated more clearly. Right now, it's very difficult to understand how the performance is changing and what to look at.
- On the same line, where do the error bars in the Figures come from? Moreover, if you have them there, can you also introduce error bars in Table 1?
Recommendation
Ask for minor revision

---

## Editorial Decision

unknown